# Coalition Building and Food Insecurity: How an Equity and Justice Framework Guided a Viable Food Assistance Network

**DOI:** 10.3390/ijerph191811666

**Published:** 2022-09-16

**Authors:** Alycia Santilli, Anna Lin-Schweitzer, Sofia I. Morales, Steve Werlin, Kim Hart, James Cramer, Jason A. Martinez, Kathleen O’Connor Duffany

**Affiliations:** 1Community Alliance for Research and Engagement (CARE), Southern Connecticut State University, New Haven, CT 06515, USA; 2Community Alliance for Research and Engagement (CARE), Yale School of Public Health, New Haven, CT 06510, USA; 3Downtown Evening Soup Kitchen, New Haven, CT 06510, USA; 4Witnesses to Hunger—New Haven Chapter, New Haven, CT, USA; 5Loaves and Fishes, New Haven, CT 06511, USA; 6United Way of Greater Waterbury, Waterbury, CT 06702, USA

**Keywords:** food insecurity, chronic disease prevention, coalition building, policy, systems, environment change

## Abstract

Food insecurity is widespread in the United States. The COVID-19 pandemic intensified the need for food assistance and created opportunities for collaboration among historically-siloed organizations. Research has demonstrated the importance of coalition building and community organizing in Policy, Systems, and Environmental (PSE) change and its potential to address equitable access to food, ultimately improving population health outcomes. In New Haven, community partners formed a coalition to address systems-level issues in the local food assistance system through the Greater New Haven Coordinated Food Assistance Network (CFAN). Organizing the development of CFAN within the framework of Collaborating for Equity and Justice (CEJ) reveals a new way of collaborating with communities for social change with an explicit focus on equity and justice. A document review exploring the initiation and growth of the network found that 165 individuals, representing 63 organizations, participated in CFAN since its inception and collaborated on 50 actions that promote food access and overall health. Eighty-one percent of these actions advanced equitable resource distribution across the food system, with forty-five percent focused on coordinating food programs to meet the needs of underserved communities. With the goal of improving access to food while addressing overall equity within the system, the authors describe CFAN as a potential community organizing model in food assistance systems.

## 1. Introduction

Food insecurity is a widespread challenge in the United States, with a national prevalence of approximately 10% [1]. This rate triples in New Haven, CT, where food insecurity impacts over 30% of adults residing in neighborhoods with the fewest economic resources and which are predominantly Black and Latinx communities [2,3]. Research shows a strong association between food insecurity and adverse health outcomes, including a significantly higher prevalence of chronic diseases, such as diabetes, hypertension, hyperlipidemia, and heart disease, as well as adverse mental health outcomes [4,5,6]. Food insecurity is associated with a 257% higher risk of anxiety and a 253% higher risk of depression [7].

During the COVID-19 pandemic, the existing inequities that lead to food insecurity were amplified. Although overall rates of food insecurity remained unchanged between 2019 and 2020, food insecurity increased significantly among certain subgroups: households that experienced pandemic-related job loss, households with children, Black adults, Hispanic/Latinx adults, and households with at least one family member not born in the US [1,8]. Corresponding mental health problems also rose among these subgroups [7,9]. While food insecurity decreased early in the pandemic after the release of stimulus checks and expanded unemployment benefits, it increased once COVID-19 relief programs expired in September 2020 [8]. Government support also left behind many households that earn above the poverty level but still cannot afford basic needs, also known as ALICE: Assisted, Limited, Income Constrained, Employed. In 2021, 41% of Connecticut families below the ALICE threshold reported that they could not afford enough food to sufficiently feed their children, and only 43% of students below the threshold qualified for government assistance [10].

Coalition building and community organizing are critical components of working toward Policy, Systems, and Environmental (PSE) changes that can ultimately lead to improved health outcomes [11]. Literature also points to the importance of coalition building in food policy and systems change, advocating for collaboration among food system actors and the need for bridging between diverse stakeholders [12,13]. While there is robust literature on the importance of coalition building for health systems change and larger food systems change, there is limited focus on food assistance systems, specifically. 

There is also limited research on COVID-related coalition building to meet food insecurity and food access needs. One study explored the experiences of community food partners and found that formal and informal social networks played a significant role in organizations’ and individuals’ abilities to adapt during the pandemic [14]. Many organizations in this study reported collaborating with similar organizations to communicate best practices and restrictions and gather information about food access needs and limitations, leading to a coordinated effort to support their local communities [14]. Those who did not have access to these networks reported that there was often redundancy and lack of coordination in their efforts [14]. Few models exist that demonstrate collaborative efforts to combat food insecurity, especially during a public health crisis.

Coalition building has the potential to address larger, systemic issues such as equitable access to food across a system. In New Haven, several community partners have formed a coalition to address systems-level issues in the food assistance system locally through the Greater New Haven Coordinated Food Assistance Network (CFAN). It is important to note that the food assistance system has historically been referred to as “emergency food assistance,” intended to provide short-term, provisional food access to people experiencing food insecurity. However, because people depend on “emergency food” on a weekly basis for long periods of time [15,16], many organizations now refer to themselves as “food assistance” providers. 

In 2018, a local health-focused organization, the Community Alliance for Research and Engagement (CARE), based at Southern Connecticut State University and Yale School of Public Health, was awarded funding through a Centers for Disease Control and Prevention program, Racial and Ethnic Approaches to Community Health (REACH). The national REACH program addresses racial and ethnic health disparities and chronic disease prevention, including access to healthy food, with a PSE emphasis. Through this funding opportunity and building on New Haven’s history of advocacy around food security in New Haven, CARE has been able to support community partners in the initiation of CFAN as an independent, inter-organizational network to address systemic issues that impact people’s ability to access food within the food assistance system.

CFAN built on a history of food assistance advocacy that took place over several years in New Haven through the Emergency Food Council (ca. 2008–2013), the New Haven Food Policy Council (commissioned by the City of New Haven in 2012), and its affiliated Food Access Working Group, which disbanded upon the creation of CFAN. Alongside and critical to these developments was the establishment of the New Haven chapter of Witnesses to Hunger (W2H) in 2013, a grassroots group of people who have experienced food insecurity in New Haven. Each of these organizations focused on policy, advocacy, leadership development, and coordination of resources and services across the local food assistance system. Moreover, each provided the platform, environment, and history of trust necessary for sustainable coalition building. Through the support of CARE and the REACH grant, and building on this foundation, CFAN convened and began its coalition-building work in April of 2019. 

The aims of this study are:(1)to detail CFAN’s initiation, development, and progress as a coalition;(2)to describe and assess CFAN’s progress within the coalition framework of Collaborating for Equity and Justice (CEJ).

## 2. Coalition Framework

The development of the CFAN coalition can be characterized within the framework of Collaborating for Equity and Justice (CEJ), which offers a way of engaging communities by facilitating cross-sector collaborations for social change, with an explicit focus on equity and justice ([17]). With the goal of improving access to food, especially healthy foods, while addressing overall equity within the system, the authors present CFAN as a potential model for community organizing around food assistance systems structured with the six CEJ principles (Table 1). CFAN explicitly addresses issues of social and economic injustice and structural racism in its vision and goals, as well as intentionally including the voices of residents who have experienced food insecurity in leadership and decision-making. Furthermore, it focuses on policy, systems, and structural change. Finally, CARE and the United Way of Greater New Haven (UWGNH), a local organization that works to create solutions to Greater New Haven’s most pressing challenges, including substantive work in food insecurity, serve as neutral conveners that secure resources, manage administrative needs, and coordinate member activities. CFAN, as a whole, consistently considers power dynamics among coalition members and the impacts of racism among residents and organizations involved. Impacts of the coalition on the food assistance system are highlighted, including how its existence at the onset of COVID-19 accelerated the community’s ability and motivation to respond to the emergency food crisis that emerged at the beginning of the pandemic and ultimately accelerated coalition development and solidification. CFAN serves as a case example of utilizing the CEJ framework for coalition building within food assistance systems and may provide a model for other regions to replicate.

## 3. Document Review Process

An iterative document review was used to detail the initiation, development, and progress of CFAN, which began in April 2019; actions and metrics are assessed through March 2022. Document reviews have proven to be a valuable method to assess the efforts of a program and to understand a program’s alignment with a given set of values or framework [11]. The document review coding team was made up of academic researchers and CFAN members, including the co-chair. Existing documents were compiled and assessed for relevance by the coding team, with CFAN members providing context. Documents were reviewed to evaluate the following elements of CFAN: vision and goals, bylaws, and guiding principles; the steering committee (agendas, attendance, meeting notes, and email templates); and the broader CFAN network (agendas, attendance, meeting notes, outreach materials, email templates, drafts of reports). These documents were used to track the number and types of actions initiated and pursued by CFAN at large; the number and types of Working Groups and Working Group actions; and the composition of CFAN (i.e., number of people attending meetings, number/types of organizations engaged, retention after COVID emergency relief period). The review also identified how CEJ principles were applied.

The document review used qualitative content analysis of the existing documents, which were coded deductively according to various aspects of the CEJ framework. Prior to the document review, the coding team developed five primary themes. The primary coder was a team member working for CARE who had not participated in CFAN. The primary coder went through approximately 20% of the documents before returning to the study team to discuss categorization. When consensus was reached and a shared understanding of the primary themes had been developed, the primary coder returned to the documents and completed the review and categorization. The results were then shared with the coding team to discuss how each item was categorized. If there was any lack of clarity, the coding team met with the full study team, which included additional CFAN members, to arrive at a consensus. Every author reviewed and approved the final categorization.

## 4. Results of Document Review

### 4.1. Overview of Initiation and Progression of CFAN

The results of the document review illustrate the initiation and progress of CFAN. CFAN began in April 2019 with a core group of partners involved in food insecurity work, including people who experienced hunger and staff from food assistance programs and backbone institutions (average attendance of the core group in initial meetings: 7.5). Initial actions included developing and refining a vision for the group, setting its goals, and planning for and organizing partners to attend its first summit to present goals to a wider group for input and to launch the network. The summit attracted 41 people representing 21 organizations in October 2019. The event provided a space to discuss the current strengths and limitations of food assistance programs in the New Haven area, network with others involved in addressing food insecurity for the purpose of sharing information and resources, and brainstorm aspirational goals for an ideal coordinated food assistance network. The summit included representatives from community food pantries and regional food banks, faith-based organizations, the City of New Haven, community centers, and nonprofit organizations. The core group continued to meet from December 2019 to March 2020 and began forming its Steering Committee while it planned the next larger network meeting for the end of March 2020. With the onset of the COVID-19 pandemic in mid-March 2020, CFAN shifted gears and intensely responded to the emergency food crisis for the next several months, leveling out to refocus on longer-term goals in September 2020. 

Over the three years that the coalition has been active, CFAN has aspired to incorporate equity and justice in all facets of its work while creating lasting systems-level change. The CEJ framework, outlined in Table 1, aligns with CFAN’s core values and is a lens through which to assess CFAN’s commitment to transformative changes in power, equity, and justice. 

### 4.2. CEJ Framework

The iterative document review confirmed that all six CEJ principles are evident in the work of CFAN. The following section outlines the ways in which CFAN aligns with the six principles of CEJ.

(a)From the outset, CFAN reflected **principle 1, explicitly addressing issues of social and economic injustice and structural racism** in the development of its vision and goals. The CFAN vision, “a unified system of food assistance that ensures equitable, dignified, and culturally appropriate access to nutritious food for all residents of Greater New Haven”, leverages language that clearly expresses this commitment, and five of its six goals focus on social justice and structural racism (Coordinated Food Assistance Network [CFAN], 2020a). Specifically, goals one, two, three, four, and six address social and economic justice with a focus on coordinating and distributing resources equitably and reducing barriers to access across the food assistance system with an emphasis on underserved communities. Goal six focuses on structural racism. See Table 2.

(b)Embodying **principle 2, in which residents have equal power in determining the coalition’s agenda and resource allocation,** CFAN is structurally organized to intentionally include residents who have experienced food insecurity. W2H has been integral in the formation of CFAN, with at least four representatives serving in all meetings during the formation of CFAN from April 2019–October 2019. Once CFAN outlined more specific guiding principles in February 2020, which were further developed in its bylaws in November 2020, it detailed that “the Steering Committee [of CFAN] will always have an equal number of food program representatives and participants of food programs, with a minimum of four from each group” [19,20]. Additionally, one of the three chairs of CFAN must be a W2H representative or a person who has experienced food insecurity. CFAN, via CARE, pays a stipend to four members of W2H for representation on CFAN, including its Steering Committee. Additionally, an agenda item has been reserved for a W2H report at all CFAN meetings so that they can update the network on its grassroots advocacy efforts and solicit support from the network.(c)Many of these actions also align with **principle 3, employing community organizing as an intentional strategy, which includes building resident leadership, specifically including residents in CFAN leadership who have experienced food insecurity.** Additionally, independent of CFAN, W2H employs a community organizing approach, which CFAN explicitly supports by collaborating closely on W2H activities with professionals from CFAN playing a supporting role where they share experience and resources but are not involved in W2H problem definition and agenda setting. Each year, W2H typically decides on a legislative agenda on state- and federal-level issues and shares the agenda and relevant actions with CFAN (if/when bills are introduced, when to call/write to legislators, etc.). CFAN also organizes to advocate for program and policy changes with our regional food bank, where most providers get food and which sets many of the rules and regulations that providers must follow, including implementation of federal regulations. When a quality improvement issue or problem is identified by providers and/or W2H members, CFAN has organized on numerous occasions to advocate for change by setting up meetings with decision-makers and writing letters. For example, after attending a training hosted by the regional food bank, W2H members reported that language used to describe populations accessing food assistance could be construed as stereotyping and disrespectful. W2H brought this issue to CFAN, where the group problem-solved and created an advocacy plan led by W2H members.(d)**Reflecting principle 4, CFAN focuses on policy, systems, and structural change,** as demonstrated in its explicit vision of creating a unified system of food assistance that ensures equitable, dignified, and culturally appropriate access to nutritious food for all residents. The CFAN goals further detail policy- and systems-level change. Actions targeting systems change are described below and outlined in Table 3. While some individual actions described may not be categorized as addressing a specific PSE change, the collective work of the coalition represents a shift toward systems and structural transformations [21,22,23,24,25,26]. For example, while the development of a food resource guide, in and of itself, may not be considered a systems change action, it is pushing forward equitable access to food across a system. Ultimately, coalition building—and the actions that emerge from coalitions—is an essential element for building PSE change [27].

Between March 2020 and March 2022, CFAN engaged in 50 actions, encompassing both emergency COVID response and larger CFAN goals. Five working groups emerged to focus on specific systems-level changes, and these working groups initiated 12 of the 50 actions [27]. At the end of May 2020, the Procurement Working Group was formed to assist pantries with procuring sufficient and appropriate foods for distribution. By August 2020, the Resources Equity Working Group and Universal Intake Working Group had formed. The Resources Equity Working Group was designed to assess gaps in New Haven’s food assistance programs and ensure equitable distribution across neighborhoods. This group was also interested in ensuring that program staff and volunteers have easy access to food assistance program information so that they could share resources with their clients. The primary goal of the Universal Intake Working Group is to streamline the intake process while satisfying requirements for The Emergency Food Assistance Program (TEFAP) across program sites to reduce barriers for people accessing food assistance and to increase the system’s overall capacity for performance measurement and improvement and system coverage. This working group has piloted an online tool in New Haven’s largest pantry and has a plan and timeline in place to roll it out to other network pantries in the next six months. The Training Working Group was formed in November 2021 in order to help organize and promote training opportunities for CFAN members that help build capacity and knowledge in areas that they have identified. After identifying overlap with the Resources Equity Working Group’s actions, it merged with the Training Working Group. A Food Gap and Summer Meals Working Group was also initiated to address food insecurity during the times that children are out of school and not receiving free breakfast and lunch. This group organizes food distribution at hub schools and organizations during these times. Specific actions conducted by working groups are summarized in Table 3, with the number of actions corresponding to each action type and CFAN Goal outlined in Table 4.
 **Impacts of COVID-19:** CFAN became a strong and trusted coalition in its formative phase by adhering to the CEJ Framework and was leveraged at the onset of the COVID-19 pandemic, dramatically impacting the course of CFAN. Being a trusted entity inclusive of multiple perspectives, CFAN was able to mobilize quickly, initiating daily meetings to respond to the crisis for seven weeks, slowly tapering to meeting 2–3 times per week before resuming its regular monthly schedule at the beginning of September 2020. These meetings attracted several new participants and organizations, and CFAN quickly pivoted to respond to the direct impacts of the COVID-19 crisis on the emergency food system that were voiced by members. Recognized as an organized entity, CFAN also began serving as a task force for the City of New Haven in early 2020. CFAN leveraged the skills and resources of its coalition members, ultimately making a substantial impact on the emergency food response in New Haven.

Notably, CFAN mobilized its network to implement the Pantry to Pantry program within two weeks of the COVID-19 shutdown. The program provided free, weekly home deliveries of food to those who were homebound and facing economic hardship due to the pandemic, most of whom had been accessing the brick-and-mortar, walk-up pantries across the system prior to March 2020 but were now being told to “stay home, stay safe”. To launch this program so quickly, CFAN’s Steering Committee developed a program model and grant proposal to present to the Yale Community for New Haven Fund, a fund set up specifically for pandemic response, and received a grant of USD 120,000 to operate the program through the summer of 2020. Ultimately, the program ran for five months and served 1310 households weekly with a total of 11,500 deliveries.

Other notable COVID-19 response actions included supporting the mobilization of a system to feed people experiencing homelessness who were temporarily moved to hotels as a result of state-mandated “shelter decompression”. CFAN members also worked with the New Haven Health Department to develop public health guidance to enable food programs to function safely during COVID-19, helped to distribute PPE, and developed a distribution system for USDA food boxes. Overall, CFAN implemented 21 COVID-specific actions over 24 months, with the majority (80.9%) being initiated between March 2020 and May 2020.

Between March 2020 and March 2022, CFAN expanded its membership to include 165 people representing 63 organizations and 10 types of organizations and engaged in 50 actions. Organization types include pantries/soup kitchens, backbone agencies, nonprofit/social service organizations, community groups, religious groups, universities, COVID response organizations, healthcare organizations, regional food banks, and other food assistance programs.

(e)**Addressing Principle 5, CFAN builds on extensive community-engaged scholarship on coalition building.** CFAN is guided by principles of coalition building and community organizing, such as the Rothman Model, which outlines three modes of community organizing practice for developing a comprehensive plan of action to mobilize communities and effect social change [28,29]. CFAN is also guided by Asset Based Community Development (ABCD), a strategy for sustainable community-driven development. ABCD builds on the assets that are already found in the community and mobilizes individuals, associations, and institutions to come together to build on their assets [30]. Specific to organizing around health-related initiatives, Minkler and Wakimoto (eds) lay out frameworks and initiatives that demonstrate the promise of community building as formidable strategies for improving health and wellbeing. This principle also calls for a good evaluation system to assess a coalition’s progress. The document review process is a first step in building out a larger evaluation plan.(f)Lastly, CFAN adheres to **principle 6** with CARE and the UWGNH serving as neutral conveners by securing resources, coordinating member activities, and managing administrative details. CARE expends resources to sustain the collaborative, including supporting W2H stipends, helping to coordinate member activities, and attending to record keeping, meeting arrangements, and distributing meeting agendas and notes. CARE and the UWGNH intentionally attempt to play a supporting role rather than taking a ‘top-down’ approach to organizing the network.

## 5. Discussion

### 5.1. Future Application of the CEJ Framework

Viewing the progress of CFAN through the framework of the “Six Principles of Collaborating for Equity and Justice” demonstrates that CFAN strives toward enhancing principles and practices in ways that will increase the likelihood of equitable and lasting systems change while also identifying gaps where CFAN can continue to improve. The document review findings have also demonstrated considerable areas of success related to completing CFAN actions that work towards its vision and goals of an equitable food assistance system.

While CFAN members regularly reference and work to address issues of social and economic injustice and structural racism (Principle 1), this principle could be better embodied with specific action items and advocacy efforts integrated throughout the coalition’s actions. CFAN already plans to offer anti-racism training opportunities to staff and volunteers at pantries and could further formalize this anti-racism commitment by advocating that the regional food bank require these ongoing trainings in order for food providers to maintain membership status with the food bank. Additionally, CFAN could more explicitly link its work to advocacy efforts around economic justice (e.g., living wage, affordable housing, jobs training).

CFAN has been successful in engaging W2H and residents who have experienced food insecurity in setting its agenda while building leadership and power, with bidirectional communication occurring between CFAN and W2H (Principles 2 and 3). Offering paid stipends has proven helpful in recognizing the work that residents put into the coalition, as well as compensating participants for their time. CFAN could further enhance the role of resident leaders through the development of more formal training opportunities and broader outreach efforts. Training could include an overview of the people and organizations who are participating in CFAN and why they are at the table, breaking down language and jargon more formally in meetings, and building skills in agenda setting and meeting facilitation—all with the goal of building W2H power as central to CFAN power. CFAN leadership has found over the last three years that client-voice plays a critical role in advocating to both public agencies and the regional food bank; more formalized trainings to develop advocacy skills such as public speaking, community organizing, and advocacy could help CFAN to meet the stated goals more fully.

Although interest in policy, systems, and structural change (Principle 4) is always present to some degree, early efforts in 2019 generally prioritized internal organizing. With the crisis brought about by the pandemic in the spring of 2020, there was little room to emphasize higher-level systems change. However, interest has grown as the severity of the crisis has waned. From the outset, CFAN included among its leadership the City of New Haven’s Food Systems Policy Director, a position created in 2015 through pre-CFAN advocacy efforts organized through the city-commissioned Food Policy Council. The importance of having a willing, capable, and engaged partner focused on policy and systems change within city government cannot be overstated. The involvement of the Food Systems Policy Director within CFAN helped foster a greater role of resident leaders with lived experience in municipal decision-making around food assistance and has given food assistance providers important insight into city processes that affect the food system (demonstrated most acutely by providers’ relationship with the Health Department in ensuring COVID-19 safety protocols and with New Haven Public Schools in organizing efforts around Summer Meals programming). From a structural perspective, CFAN could implement a working group focused on policy change to develop a more sophisticated strategy for both the immediate and long term.

Expanding the evaluation of the CFAN coalition approach (Principle 5) presents ample opportunity. The idea of an evaluation of CFAN has recently organically risen within the network independent of the document review process. CARE, one of the main conveners of CFAN, focuses much of its health equity work on research and evaluation and has supported activities with formative and process evaluations and needs assessments, but has not yet conducted a formal evaluation of CFAN due to the timing of the pandemic and potential burden of introducing an extensive evaluation in the midst of a crisis. However, the opportunity to conduct the document review and assess CFAN’s work within the CEJ framework has provided a solid foundation for taking the next step in evaluation, particularly as CFAN moves toward formalizing internal operations and seeking long-term funding streams in order to strive for an even greater impact.

Finally, in order to fully embody an emphasis on facilitating structures that promote equity and justice (Principle 6), CARE and UWGNH, as conveners, can focus on being more vigilant of power dynamics among coalition members and more intentional about power imbalances and the impacts of racism among residents and organizations involved in CFAN. While CFAN has begun writing bylaws and has developed some internal governing rules, they currently do not outline the terms and election of Steering Committee members and co-chairs, general network membership, or clear means for directing policy internally. The lack of structure has the potential to allow for underrepresented voices to be heard; however, it also risks enabling the loudest voice in the room, presenting a dynamic that favors those who have traditionally been in positions of power. The role of the conveners in facilitating this process could be enhanced and formalized.

Throughout its short history thus far, CFAN has lacked sustainable, long-term funding streams. Now, however, CFAN has come to an inflection point: having demonstrated broad community buy-in from a variety of members, including people with lived experience, direct service providers, and supporting organizations, CFAN has the potential for much greater impact, provided that the work is carried out in a deliberate, strategic, and organized manner. A steady source of funding would provide for greater sustainability and increased capacity through a dedicated staff position. Dedicating staff resources to CFAN’s internal operations would help keep the coalition coordinated, move toward fulfilling its vision and goals, and implement strategies that more fully embody the CEJ framework.

### 5.2. Recommendations for Coalition-Building for Equity and Justice

Other communities across the United States can apply these lessons learned to achieve greater equity and justice as directed by people experiencing food insecurity. A well-built coalition can withstand—and even become stronger—when forced to manage an emergency response in the food assistance system, as demonstrated by CFAN’s ability to strengthen during the COVID crisis. Similar to a previous study, CFAN demonstrated a high level of collaboration in sharing resources and communicating best practices, leading to a coordinated effort to support the New Haven community ([14]). With a continued and sustained intention and focus on implementing its work through the CEJ framework, Principles of Collaborating for Equity and Justice, CFAN is a promising model that has the potential for leading transformational changes within food assistance systems. For communities looking to replicate this model within food assistance systems and more broadly, recommendations to support similar work are summarized in Table 5.

Furthermore, the recommendations can be broadly applied to coalition building across many different social justice issues. It is particularly important to ensure representation of people with lived experiences in building any coalition, including ensuring representation within coalition leadership. People with lived experiences should help direct the focus, goals, and actions of coalitions. Ensuring diversity of representation across all partners strengthens coalitions. Establishing trust among members is, perhaps, one of the most important aspects of coalition building. One strategy for achieving trust is developing goals with the coalition as a whole (versus any type of “top-down” approach) in order to ensure buy-in. Then, members can, ideally, come together to initiate specific actions toward those goals, and through this process of action and goal achievement, trust continues to grow as members experience the benefits of working together on solutions. Working groups or committees can be formed to help attain goals but should develop organically (again, avoiding any top-down approaches) and with consideration of the coalition’s capacity to undertake working group activities (setting agendas, organizing and facilitating meetings, and following through with action steps). As goals are achieved, more formal governance and infrastructure can be put into place to provide support and guide the overall direction and decision-making of the coalition. Formal governance development does not necessarily need to be the first step in forming a coalition. Finally, centering equity and justice with intentional actions and consistently assessing and reflecting on related progress is vital to truly achieving social change. 

## 6. Conclusions

This study describes an equity-focused process for building a coalition to address food insecurity and access to healthy foods. The Principles of Collaborating for Equity and Justice is an essential framework for coalitions to utilize as a guide for creating equitable partnerships and more just societies. This study underscores the overall importance of coalition building for policy, system, and environmental change. 

## Figures and Tables

**Table 1 ijerph-19-11666-t001:** Principles for Collaborating for Equity and Justice [17].

**Principle 1**	Explicitly address issues of social and economic injustice and structural racism.
**Principle 2**	Employ a community development approach in which residents have equal power in determining the coalition’s or collaborative’s agenda and resource allocation.
**Principle 3**	Employ community organizing as an intentional strategy and as part of the process. Work to build resident leadership and power.
**Principle 4**	Focus on policy, systems, and structural change.
**Principle 5**	Build on the extensive community-engaged scholarship and research over the last four decades that show what works, acknowledge the complexities, and that evaluate appropriately.
**Principle 6**	Construct core functions for the collaborative based on equity and justice that provide basic facilitating structures and build member ownership and leadership.

**Table 2 ijerph-19-11666-t002:** Coordinated Food Assistance Network: Vision and Goals, updated May 2020 [18].

**Vision**: A unified system of food assistance that ensures equitable, dignified, and culturally appropriate access to nutritious food for all residents of Greater New Haven.	
**Goal 1**	Support the coordination of neighborhood-based food programs in New Haven, aligning locations and hours. Ensure that needs are met, emphasizing underserved communities and addressing gaps.
**Goal 2**	Create universal eligibility requirements for all food assistance programs to reduce barriers to access, commensurate with available resources.
**Goal 3**	Streamline and coordinate a universal intake process across sites to enable data collection that informs service improvement, performance measurement, and system coverage.
**Goal 4**	Provide means and support system for ensuring that food, supplies, volunteers, transportation support services, training resources, and other resources can be pooled or shared when appropriate. Ensure resources are distributed equitably across the food system. Especially, ensure formal structures are in place for equitably responding to emergencies and disasters, both acute and chronic.
**Goal 5**	Promote quality standards across the system for nutrition and food safety.
**Goal 6**	Provide effective and systematic training on cultural humility to ensure staff and volunteers of food assistance programs treat all guests with dignity, respect, and love.

**Table 3 ijerph-19-11666-t003:** Actions conducted by CFAN from April 2019 to March 2022.

Actions	Type	Goal #	Initiation Date
Networked emergency food system initiated	core CFAN action	0 (vision)	April 2019
Develop CFAN Visions and Goals	core CFAN action	0 (vision)	April–September 2019
Implement Supporting Wellness at Pantries (SWAP), a program to increase pantries’ inventories of healthy food	core CFAN action	5	May 2019
Plan and host first CFAN summit (28 October 2019)	core CFAN action	0 (vision)	July–October 2019
Create and distribute “Food Resource Guide”	core CFAN action	1	September 2019
Organize Give Healthy digital food drives to collect healthy foods for pantries	core CFAN action	5	October 2019
Distribute food to people experiencing homelessness living in hotels during the pandemic	COVID	1, 4	March 2020
Soup kitchens switch to “grab and go” model for serving food	COVID	1, 4	March 2020
Coordinate delivery of prepared food	COVID	1, 4	March 2020
Coordinate Pantry to Pantry grocery/food delivery program	COVID	1, 4	March 2020
Secure grant funding for Pantry to Pantry program	COVID	1, 4	April 2020
Begin school meal distribution	COVID	1, 4	March 2020
Coordinate people interested in food sharing	core CFAN action	4, 5	March 2020
Coordinate “grab and go” program for senior centers	COVID	1, 4	March 2020
Create “COVID-19 Food Resource Guide”	COVID	1, 4	March 2020
Create and distribute COVID safety guidelines for food distribution	COVID	4	March 2020
Plan for and initiate USDA food box distribution (start May 2020)	COVID	1, 4	April 2020
Coordinate pop-up food pantries	COVID	1, 4	April 2020
Collect data on clients attending pop-up food pantries to assess needs for future pop-ups	COVID	1, 4	April 2020
COVID response logistics group	COVID	4	April 2020
PPE procurement and warehouse	COVID	4	April 2020
COVID Pop-up Center—resources for un-housed population	COVID	4	May 2020
Support reopening of closed pantries	COVID	4	May 2020
Create Procurement Working Group	COVID	4	May 2020
Bulk/distributor food purchasing	COVID	4	May 2020
Survey pantries on challenges of scaling up services during COVID and create/present report	COVID	1, 4	July 2020
Create Resources Equity Working Group	core CFAN action	4	August 2020
Create Universal Intake Working Group	core CFAN action	2, 3	August 2020
Thanksgiving list of pantries and organizations distributing frozen turkeys (annual)	core CFAN action	4	November 2020
Coordinate COVID vaccine outreach	COVID	4	February 2021
Pilot universal intake process at largest pantry	Working Group (Universal Intake)	3	March 2021
Design custom web app to support intake process	Working Group (Universal Intake)	3	May 2021
Community Food Hub/Fill the Shelves Initiative: sharing refrigeration space between pantries	core CFAN action	1, 4	June 2021
Host second CFAN summit	core CFAN action	0 (vision)	July 2021
Assess resources and determine that 211 has appropriate infrastructure; merge with Training Work Group to host 211 trainings	Working Group (Resource Equity)	1, 4	September 2021
Design Google widget to allow advanced search for pantries, including hours of operation and details	Working Group (Resource Equity)	1, 4	October 2021
Coordinate Flu vaccination clinics at pantries/soup kitchens	core CFAN action	4	November 2021
Create Training Working Group	core CFAN action	4, 6	November 2021
Distribute “CFAN Resource Guide”	Working Group (NHPS Food Gap)	1, 4	December 2021
Distribute “CFAN Holiday meals guide”	Working Group (NHPS Food Gap)	1, 4	December 2021
Create “Food Gap 1-pager”	Working Group (NHPS Food Gap)	1, 4	December 2021
Create “CFAN Training Opportunities Shared Document”	Working Group (Training Opportunities)	4	December 2021
New Haven Food Gap Initiative: coordinate food distribution during holidays/vacations	Working Group (NHPS Food Gap)	1, 4	January 2022
Create Neighborhood Pop-Up List	Working Group (NHPS Food Gap)	1, 4	January 2022
Distribute KN95 masks at pantries	COVID	4	January 2022
Advocate for improved trainings that incorporate principles of cultural humility	core CFAN action	6	January 2022
Advocate for the regional food bank to implement a food pantry participant bill of rights that incorporates cultural humility in practices at pantries	core CFAN action	6	January 2022
“February Gap” food distribution	Working Group (NHPS Food Gap)	1, 4	February 2022
Create “Meal Gap” subgroup of NHPS Food Task Force	core CFAN action	1, 4	February 2022
Coordinate 211 trainings for CFAN members	Working Group (Training Opportunities)	4	March 2022

**Table 4 ijerph-19-11666-t004:** Number of actions corresponding to each CFAN Goal and action type.

	Number of Corresponding Actions
**Goals**	
0 (vision)	4
Goal 1	23
Goal 2	1
Goal 3	3
Goal 4	38
Goal 5	3
Goal 6	3
**Action Type**	
Core CFAN action	17
COVID	21
Working Group (Resource Equity)	2
Working Group (Universal Intake)	2
Working Group (Training Opportunities)	2
Working Group (NHPS Food Gap)	6
**Total number of actions**	50

**Table 5 ijerph-19-11666-t005:** Recommendations for coalition-building for equity and justice.

Representation	Buy-In	Shared Values
Center the voices of those who have been **historically underrepresented** in your community. Leadership is important in keeping the coalition moving forward, but **leaders should be representative of the members and serve at their direction**; tri-chairs infrastructure can work great, especially when they get along well. Be inclusive of all community partners and residents, and be honest in identifying partners and especially residents with **lived experience**.	Do not get hung up on formality in structure before you have **buy-in and trust**; based on readiness, creating more formality in structure can be a means for ensuring inclusion. Be aware of, and address the needs of all partners and residents. **Not everyone comes to the table for the same reason**. Ensure buy-in from all partners through **goal development and selection of activities**. If members stop showing up for meetings, they’re probably not on board. Be patient with your coalition, and **meet your members where they are at**—but avoid stagnation. **Allow working groups to form organically based on members’ interests and identified needs** (vs. predetermined committee infrastructure). This allows members to take ownership of the direction of the larger network and creates buy-in of specific issues to address.	Become familiar with the **CEJ principles**. Collaboratively develop goals with the coalition to **establish a common purpose** and always keep them front and center. **Never take your coalition for granted**. It is hard work to keep it together, but it undeniably makes for a better network of services, clearer focus on equity, and ultimately, it builds a stronger community.

## Data Availability

The data presented in this study are available on request from the corresponding author.

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
