# Peer review of "Coalition Building and Food Insecurity: How an Equity and Justice Framework Guided a Viable Food Assistance Network"

_ijerph, 2022, doi:10.3390/ijerph191811666_

Round 1

Reviewer 1 Report

The article describes laudable efforts to address food insecurity in New Haven Connecticut using an equity lens and a comparison of those efforts to the Collaborating for Equity and Justice Framework. The description of coalition work is likely of interest to readers and is important in order to advance equity in PSE change and coalition efforts. 

However, the methods section of the paper lacks sufficient detail. The study does not describe the process of document review for readers, nor is the concept of a document review defined. The authors need to provide citations for the methodology of the review, describe which authors were involved in the review, how authors decided which aspects of the documents fell into various aspects of the CEJ framework, and how discrepancies were resolved. It is possible it may be more accurate to describe this process as a qualitative content analysis of the existing documents which were coded deductively according to various aspects of the CEJ framework. If so, the authors may wish to cite papers describing content analysis methods. 

The results section may also be more appropriately presented as a qualitative case study or commentary, because the authors describe work that CFAN has done and how it fits into CEJ principles, but there is no further analysis or presentation of data.

The authors spend a majority of the discussion section discussing how these results affect the work of CFAN. I would encourage the authors to expand upon the last paragraph and recommendations in table 5 in order to provide additional information that may be useful to practitioners in other areas. Otherwise, much of this section reads as  an evaluation report that would be provided to CFAN and not something that would be useful to readers outside of CFAN. 

The paper does present information that is important for readers and could help advance equity and social justice in coalition work, but the authors will need to revise the paper to include more information about their methods and to demonstrate how their results are useful to readers who did not work with CFAN.

Reviewer 2 Report

The authors explore the inception, development, and progress of CFAN which is based on the framework of the “Six Principles of Collaborating for Equity and Justice”.

The article is relevant and timely. The paper is straightforward, well-written and well structured. Tables are informative and clearly presented. The discussion is clear and comprehensive. However, I have several minor issues which should be addressed prior to publication:

Broad comments:

·    I have two issues with the introduction. First, I cannot see the relation between the last paragraph and the rest of the introduction. How are CARE and CFAN related? And second, the authors do not include an aim. I would suggest further elaborating both points, in other words, clarifying the relation between the last paragraph’s content and the rest of the section and including the paper’s objective.

·       The United Way of Greater New Haven (UWGNH) (line 119) has not been mentioned until now. I would suggest properly introducing it.

·       Sections 4.1/4.2: I would suggest including notes in the body of the text to explain what Table 1 shows wherever pertinent (as done for Table 2 in line 178 or Table 3 in line 218).

·       Section “4.2. CEJ Framework”: I would suggest including an introductory note. It could be the sentence “The following section outlines the ways in which CFAN aligns with the six principles of CEJ”, currently written in lines 165-166.

·       Are there any other similar experiences? I miss some discussion related to this issue.

Specific comments:

·   Line 196: The text reads “Many of these activities …” I would suggest specifying which activities the authors refer to.

·       Line 230: The Procurement Working Group is not included in Table 3.

·       Table 3: Goal “0” of action “Host second CFAN summit” is written as “0” when in previous cases is written as “0 (vision)”. Just in case the authors want to change it.

·       Table 4: According to table 3, the number of corresponding actions for Goal 1 are 23, not 22.

·       Table 4: According to table 3, the number of corresponding actions for Goal 4 are 38, not 37.

·       Table 4: According to table 3, the number of corresponding actions for Core CFAN action are 17, not 18. The total number of actions would not be 50. Despite this, Table 3 shows 50 actions.

Round 2

Reviewer 1 Report

Thank you for addressing my comments related to the methods and discussions sections. The paper now provides sufficient detail and perspective to be of value and interest to readers and I recommend that it be accepted for publication.